# Changes in Levels of Serum Cytokines and Chemokines in Perforated Appendicitis in Children

**DOI:** 10.3390/ijms25116076

**Published:** 2024-05-31

**Authors:** Wen-Ya Lin, En-Pei Lee, Chun-Yu Chen, Bei-Cyuan Guo, Mao-Jen Lin, Han-Ping Wu

**Affiliations:** 1Department of Pediatrics, Taichung Veterans General Hospital, Taichung 40705, Taiwan; wylin002@gmail.com; 2Pediatric Sepsis Study Group, Division of Pediatric Critical Care Medicine, Department of Pediatrics, Chang Gung Memorial Hospital at Linko, Kweishan, Taoyuan 33305, Taiwan; pilichrislnp@gmail.com; 3College of Medicine, Chang Gung University, Taoyuan 33302, Taiwan; 4Department of Emergency Medicine, Tungs’ Taichung Metro Harbor Hospital, Taichung 43503, Taiwan; yoyo116984@gmail.com; 5Department of Nursing, Jen-Teh Junior College of Medicine, Nursing and Management, Miaoli 356006, Taiwan; 6Department of Pediatrics, National Cheng Kung University Hospital, College of Medicine, National Cheng Kung University, Tainan 356006, Taiwan; gbc628@gmail.com; 7Department of Medicine, Taichung Tzu Chi Hospital, The Buddhist Tzu Chi Medical Foundation, Taichung 427213, Taiwan; 8Department of Medicine, College of Medicine, Tzu Chi University, Hualien 970374, Taiwan; 9Department of Pediatrics, Chiayi Chang Gung Memorial Hospital, Chiayi 61363, Taiwan

**Keywords:** perforated appendicitis, cytokines, chemokines, children

## Abstract

Appendicitis is primarily diagnosed based on intraoperative or histopathological findings, and few studies have explored pre-operative markers of a perforated appendix. This study aimed to identify systemic biomarkers to predict pediatric appendicitis at various time points. The study group comprised pediatric patients with clinically suspected appendicitis between 2016 and 2019. Pre-surgical serum interleukin-6 (IL-6), tumor necrosis factor-alpha (TNF-α), intercellular cell-adhesion molecule-1 (ICAM-1), and endothelial selectin (E-selectin) levels were tested from day 1 to day 3 of the disease course. The biomarker values were analyzed and compared between children with normal appendices and appendicitis and those with perforated appendicitis (PA) and non-perforated appendicitis. Among 226 pediatric patients, 106 had non-perforated appendicitis, 102 had PA, and 18 had normal appendices. The levels of all serum proinflammatory biomarkers were elevated in children with acute appendicitis compared with those in children with normal appendices. In addition, the serum IL-6 and TNF-α levels in children with PA were significantly higher, with an elevation in TNF-α levels from days 1 and 2. In addition, serum IL-6 levels increased significantly from days 2 and 3 (both *p* < 0.05). Serum ICAM-1 and E-selectin levels were elevated in the PA group, with consistently elevated levels within the first three days of admission (all *p* < 0.05). These results indicate that increased serum levels of proinflammatory biomarkers including IL-6, TNF-α, ICAM-1, and E-selectin could be used as parameters in the prediction and early diagnosis of acute appendicitis, especially in children with PA.

## 1. Introduction

Acute appendicitis is a common surgical emergency, accounting for 1–8% of children with acute abdominal pain presenting in the emergency room [1]. Appendicitis is typically treated with surgical appendectomy; however, an increasing number of studies have revealed that nonsurgical treatment is a safe and effective option in selected appendicitis patients, while surgery is reserved for patients with perforated appendices [2].

The proposed pathogenesis of appendicitis includes initial intraluminal obstruction by an appendicolith, lymphoid hyperplasia, or fibrous band, followed by bacterial invasion and resulting inflammation [3,4]. However, no definite obstructive lesions can be identified in the majority of appendicitis cases [5]. Newer alternative explanations have proposed initial inflammation, increased intraluminal pressure, vascular compromise, and appendiceal obstruction as drivers of appendicitis. This inflammatory process could be triggered by a viral infection followed by secondary bacterial invasion. Other factors such as diet, environmental factors, trauma, and genetic variations have also been shown to influence the development of appendicitis [4,6].

For many years, untreated appendicitis has been considered a progressive disease that will result in eventual perforation, imposing a risk of morbidity and mortality without early surgical intervention. This has prompted vigorous interventions aimed at early appendicitis diagnosis and immediate surgery in all patients with suspected appendicitis. With this traditional practice of rapid surgical intervention for all patients with suspicion of appendicitis, negative appendectomies may be found. However, negative appendectomies do not reduce the rate of appendectomies for perforated appendicitis. In addition, they also confer a substantial clinical and financial burden [7]. Negative appendectomies are also not without risks. The risk of re-admission or complications in children with negative appendectomies is comparable to that in children with appendicectomies for confirmed appendicitis [8]. These risks may be potentially avoided with better diagnostic tools, higher risk identification, and more accurate treatment advice.

Current studies have suggested that appendicitis has distinct clinical presentations, including simple appendicitis (SA), gangrenous appendicitis (GA), and perforated appendicitis (PA), while research has shown that progression from SA to PA is not inevitable in all patients [9]. As such, the treatments for the various clinical presentations of appendicitis may differ from the surgery advised for children with perforated appendicitis. More effort should thus be made to achieve accurate pre-operative differentiation and diagnosis of perforated appendices.

This study aimed to identify the serum cytokine and chemokine profiles of children with PA to identify biomarkers to facilitate pre-operative diagnosis.

## 2. Results

During the study period, a total of 226 children with RLQ abdominal pain admitted to the ED were enrolled in this study. Among the 226 children who underwent surgical intervention, 70 had SA, 36 had GA, 102 had PA, and 18 had normal appendices. The study group comprised 135 males (59.7%) and 91 females (40.3%), with a mean age of 10.0 ± 4.5 years. In this study, diagnoses were confirmed by histological findings in the surgical specimens, and a higher proportion of PA (45%) was found. The patients’ characteristics in two groups, non-PA and PA, are shown in Table 1. The levels of all serum proinflammatory biomarkers were elevated in children with acute appendicitis compared with those in children with normal appendices. In terms of clinical symptoms and signs, fever, migration pain, anorexia, nausea/vomiting, local tenderness over the RLQ, and rebound pain were significant parameters differentiating between children with PA and those with non-perforated appendices. Fever, local tenderness over the RLQ, and rebound pain were more common in patients with PA (all *p* < 0.05), while migration pain, anorexia, and nausea/vomiting were more common in patients with non-perforated appendices (all *p* < 0.05). In addition, in routine laboratory tests, only the CRP level was a significant parameter differentiating between children with PA and those with non-perforated appendices. Blood CRP levels were higher in children with PA than those in children with non-perforated appendices (188.3 ± 105.8 mg/L vs. 125 ± 95.6 mg/L; *p* < 0.05). The length of stay in hospital was also longer in children with PA than in children with non-perforated appendicitis (15.4 ± 6.5 days vs. 7.2 ± 4 days; *p* < 0.05).

Further analysis of the levels of serum cytokines, including IL-6 and TNF-α, in children with non-perforated appendicitis and PA is shown in Table 2 and Table 3, respectively. Serum IL-6 and TNF-α levels in the PA group were higher than those in the non-perforated appendicitis group. A higher level of TNF-α in the PA group was identified on day 1 and day 2, and an increased IL-6 level was found on day 2 and day 3 of admission (both *p* < 0.05). In addition, as shown in Table 4 and Table 5, the levels of serum chemokines, including ICAM-1 and E-selectin, were both highly elevated in the PA group compared with those in the non-perforated appendicitis group, with a trend toward consistent elevations in serum ICAM-1 and E-selectin levels from day 1 to day 3 (all *p* < 0.05).

## 3. Discussion

In children with acute appendicitis, the surgical approach is not without risks or complications [10,11]. Perforated appendicitis can occur in 17% to 32% of adult patients [12] and 25.9% of children [13]. The proportion of PA varies between different studies. This might be due to the different methods utilized for the diagnosis of PA. As such, treatment should be tailored individually, with surgical management provided for children with PA [2]. Multiple studies have demonstrated antibiotic-first treatment for selected SA to be a safe and effective option [1,14,15] Surgery is recommended in the presence of SA with appendicoliths due to a high failure rate [16,17]. In contrast, surgery remains the treatment of choice for children with PA, with early appendectomy recommended to reduce complications [18].

CT and ultrasonography have become important tools for diagnosing appendicitis. However, these procedures are not always readily available in all healthcare settings. In addition, their costs remain high. In Taiwan, the cost of a CT is higher than that of admission to the observation unit in the ED. Moreover, in some pediatric patients, abdominal CT or ultrasonography may show equivocal findings. Children with equivocal CT findings cannot be discharged home and need to be admitted for further survey and observation. Therefore, we believe that measuring serum levels of proinflammatory biomarkers in patients under short-term observation is a potentially helpful method of diagnosing equivocal appendicitis [19].

Acute appendicitis is an inflammatory process mediated by various cytokines and chemokines. Appendicitis is initiated by inflammation involving the recruitment of leukocytes to sites of inflammation. Serum proinflammatory biomarkers could be initially used in the pre-operative prediction of perforated appendicitis in children. Neutrophils have a number of cytokine receptors, including the type I cytokine receptor and members of the TNF receptor family responsible for IL-6 and TNF-α recognition. IL-6 activates neutrophils and regulates the inflammatory response [20]. Leucocyte recruitment is critical in the inflammatory process, which is a multistep process regulated by critical cytokines and chemokines at various stages [21]. Different families of adhesion molecules, including ICAM-1 and E-selectin, are critical for leukocyte recruitment (rolling, firm adhesion, and transendothelial migration) to the activated endothelium. E-selectin is important in the rolling stage. The synthesis and expression of E-selectin on endothelial cells are induced by TNF-α. ICAM-1, which is located in the endothelium and monocytes, is important for adherence and migration. TNF-α promotes the leukocyte adhesion process more slowly, enhancing the transcription-dependent expression of adhesion molecules, further augmenting the E-selectin and ICAM response [22]. The levels of these pro-inflammatory mediators (IL-6, TNF-α, ICAM-1, and E-selectin) should be increased during the inflammatory process and were selected accordingly.

As we know, ICAM-1 is a member of the immunoglobulin supergene family, which is responsible for the adhesion of lymphocytes, granulocytes, and monocytes to activated endothelial cells. E-selectin, a member of the selectin family, may be involved in the initial binding of neutrophils, T-lymphocytes, and monocytes to endothelial cells. The up-regulation of cell adhesion molecules could be mediated by the release of proinflammatory cytokines. The expression of E-selectin may correlate with the extent and stop of the inflammatory reaction. Within 4 h after cytokine stimulation, E-selectin may increase to a peak of expression on endothelial cells. In some studies, the researchers reported that E-selectin was the first inducible cell adhesion molecule expressed in early appendicitis and was highly expressed in advanced ulcerophlegmonous appendicitis. In addition, they stated ICAM-1 showed maximal expression in phlegmonous appendicitis and remained at this level until the stage of ulcerophlegmonous appendicitis, followed by a decline in gangrenous appendicitis, and they also showed a steady increase in E-selectin expression up to the stage of ulcerophlegmonous appendicitis (24–48 h) [23].

Moreover, IL-6 is considered an early biomarker of tissue damage and the inflammatory response. It plays an important role in the induction of an acute-phase response and anti-inflammatory activity. Appendicitis is a common surgical problem associated with a systemic inflammatory response. In addition, the levels of septum IL-6 may increase after surgery based on the degree of surgical stress. The diagnostic value of IL-6 and its capacity to indicate the extent of appendicitis has been evaluated [24]. Systematic reviews and studies revealed a possible elevation of IL-6 levels in children with complicated appendicitis [25,26]. A prospective observational study was performed to verify the diagnostic role of IL-6 in distinguishing between uncomplicated and complicated pediatric appendicitis. The results disclosed significantly elevated IL-6 levels in children with complicated appendicitis [27]. This was echoed by a retrospective study of children with acute appendicitis, where IL-6 and other biomarkers such as CRP, immunoglobulin E (IgE), and interleukin-13 (IL-13) were found to be independent risk factors for complicated appendicitis (*p* < 0.05) [28].

E-selectin is expressed on microvascular endothelial cells in response to inflammatory mediators such as IL-4 and TNF. The expression of E-selectin on cultured endothelial cells peaks within 4 to 6 h of cytokine stimulation. E-selectin may contribute to the adhesion of monocytes and a subpopulation of T lymphocytes in vitro, which are to be expressed on endothelial cells at sites of mononuclear leukocyte infiltration [29]. Tissue E-selectin and circulating E-selectin levels increase during active inflammatory bowel disease [30].

In addition, PA represents a more severe clinical disease and an increased degree of inflammation. Perforation, or a lack thereof, may influence surgeons to choose the indication of operation and the type of appendectomy. Perforation could be correlated with increased levels of proinflammatory cytokines and chemokines. Thus, the roles of IL-6, TNF-α, ICAM-1, and E-selectin in predicting pre-operative PA were investigated. In this study, the 3-day trends in cytokine and chemokine level changes in children with PA and non-perforated appendices were observed. Serum TNF-α levels increased from days 1 and 2, and a higher IL-6 level was found from days 2 and 3 in children with PA. The IL-6 level was nine-fold higher in the PA group on day 3 of the disease. ICAM-1 and E-selectin levels were consistently and significantly elevated from days 1 to 3.

Appendicitis is mostly classified based on intraoperative or histopathological findings [31]. Fewer studies have attempted to identify early markers of perforated appendices. Gurien et al. found that the WBC count at the time of admission was a significant predictor of PA in children [32]. Pre-operative serum levels of IL-6 and IL-8 and postoperative IL-6 serum levels remained significantly higher in patients with PA [33]. In this study, children with PA clearly exhibited higher levels of IL-6, TNF-α, ICAM-1, and E-selectin, which are critical players in the inflammatory process and leukocyte recruitment by endothelial cells. An analysis of multiple inflammatory markers from day 1 to day 3 in children with perforated appendicitis was presented in this study. This is unique among existing studies. The levels of certain cytokines such as ICAM-1 and E-selectin were consistently elevated within the first 3 days of presentation. TNF-α and IL-6 demonstrated increased levels from days 1 and 2, and days 2 and 3, respectively. This is clinically useful and may offer more assistance for early diagnosis.

In current clinical practice, appendectomy is the predominant treatment for acute appendicitis. However, if appendicitis is diagnosed early, it may be useful to treat it with antibiotics only rather than immediate surgery. Clinical decision making can be difficult when primary care physicians are faced with a choice between conservative and surgical treatment. This decision depends on the severity of the appendix inflammation and the serial pathogenesis of the appendicitis. Some patients with appendicitis may not exhibit a typical clinical presentation and clinical distinctions between perforated and non-perforated appendices may be challenging. The pathogenesis of appendicitis and related inflammatory biomarkers may thus require further investigation [34].

The clinical focus of appendicitis should shift toward the early pre-operative identification of children with PA and timely surgical intervention. Accurate early specific diagnoses bear more weight than rapid diagnoses. Increased serum levels of IL-6, TNF-α, ICAM-1, and E-selectin may be useful in early PA diagnosis, optimizing combined decision making for surgical treatment. Other topics of interest include possible medical prevention. Biological agents, such as anti-TNF and anti-IL-6 are available and widely used to treat other clinical diseases. In children with PA, the possible role of these biological agents in the early regulation of the inflammatory response, with the goal of preventing clinical PA, warrants further study.

## 4. Materials and Methods

### 4.1. Patient Population

This prospective study enrolled children aged 4 to 18 years who underwent surgery for suspected appendicitis in the pediatric emergency department (ED) between 2016 and 2019. All of these patients presented with symptoms and signs suggestive of appendicitis, including migration of pain, anorexia, nausea, rebound tenderness in the right lower quadrant (RLQ) of the abdomen, pyrexia, and tenderness over the right iliac fossa. The exclusion criteria included a symptom duration of >3 days and patients under 4 years of age. This was due to the equivocal presentation of clinical symptoms and signs of appendicitis in these conditions. The definitive diagnosis of appendicitis was confirmed by histopathological examination of the excised appendix, with transmural granulocyte invasion as the histopathological criterion. In addition, a patient was defined as having a normal appendix when a nonsurgical patient discharged from the ED was followed up by a telephone interview 2 weeks after the index visit to confirm that the diagnosis of appendicitis could be ruled out or when a histologically uninflamed appendix was found in a patient who had undergone surgery.

The study protocol was approved by the Institutional Review Board of the Chang Gung Memorial Hospital (IRB no.:1049627A3) and the relevant Ethics Committee and was conducted in accordance with the guidelines and the Declaration of Helsinki. Informed consent was obtained from the parents and/or the legal guardians of the children who participated in this study. All methods were performed in accordance with relevant protocols.

### 4.2. Study Design

This was a prospective observational study and was not intended to influence the indications and timing of the operative approach. Faculty physicians completed standardized patient data sheets on patients' presentations at the hospital. The age, sex, temperature, and clinical symptoms and signs of the enrolled children were recorded, and blood samples for routine laboratory tests and cytokine and chemokine analyses were collected between days 1 and 3 of admission. We identified the duration within 24 h as day 1, 24 to 48 h as day 2, and 48 to 72 h as day 3 in our study.

The total white blood cell counts(WBCs) and percentage neutrophil counts were measured by an automated five-part leukocyte differential count hematology analyzer (Cell-Dyn 4000R System, Abbot Laboratories, Abbot Park, Chicago, IL, USA). A WBC count above 10,000/mm^3^ and a neutrophil count greater than 80% were considered abnormal. The non-segmented band counts were carried out manually by counting 100 consecutive cells in a peripheral blood film. A manual band count greater than 5% was considered abnormal. The concentration of C-reactive protein (CRP) in serum was measured by immunoturbidimetry (Beckman Coulter, Fullerton, Calif). A CRP concentration above the detection level of 8.0 mg/L was taken to be elevated.

The measured cytokines included interleukin-6 (IL-6) and tumor necrosis factor-alpha (TNF-α). Chemokines, including intercellular cell adhesion molecule-1 (ICAM-1) and endothelial selectin (E-selectin), were measured with the chemokine biochip array using the semi-automated Evidence Investigator (Randox Laboratories Ltd., Crumlin, Co., software (Antrim, UK)). These serum cytokines and chemokines were all sampled prior to surgery. The excised surgical specimens were histologically divided into simple appendicitis (SA), gangrenous appendicitis (GA), and perforated appendicitis (PA). SA was defined as neutrophil infiltration of the mucosa, submucosa, or muscular is propria. GA included specimens showing transmural inflammation, necrosis, and mucosal ulceration. PA was defined as extensive transmural inflammation with perforation.

### 4.3. Statistics

The Mann–Whitney U test, chi-square test, one-way analysis of variance (ANOVA), and multivariate logistic regression analysis were used. The values are presented as means ± standard deviation (SD). The differences between groups are presented as 95% confidence intervals (CIs), and statistical significance was defined at the *p* < 0.05 level. All statistical analyses were conducted using the IBM SPSS Statistics software (version 22.0; SPSS Inc., Chicago, IL, USA).

### 4.4. Limitation

This study had only a small sample size and was conducted at a single center, which may have introduced bias. In addition, although the tests may not provide potential added value for diagnostic purposes now, we believe, in the future, much more advanced techniques could be combined to perform the analysis of biomarkers to predict appendicitis more effectively and easily based on our findings.

## 5. Conclusions

Serum proinflammatory biomarkers, including elevated serum IL-6, TNF-α, ICAM-1, and E-selectin levels, could have a role in predicting perforated appendicitis in children admitted to the ED.

## Figures and Tables

**Table 1 ijms-25-06076-t001:** Comparison of demographic characteristics between children with non-perforated appendices and perforated appendices.

Variable	Non-Perforated Appendices (*n* = 106)	Perforated Appendices (*n* = 102)	*p*-Value
Age (year)	9.8 ± 3.7	10.1 ± 3.9	0.570
Gender (male)	64 (60.4%)	51 (50.0%)	0.132
Clinical symptoms			
Fever	49 (46.2%)	58 (56.9%)	0.125
Migration pain	65 (61.3%)	42 (41.2%)	0.004
Anorexia	85 (80.2%)	48 (47.1%)	<0.001
Nausea/vomiting	61 (57.5%)	50 (49.0%)	0.218
Local tenderness over RLQ	104 (98.1%)	102 (100.0%)	0.163
Rebound pain	88 (83.0%)	96 (94.1%)	0.012
Laboratory examination			
WBC, 10^3^ uL	16.8 ± 5.8	16.4 ± 6.4	0.637
Neutrophils, %	82.0 ± 10.5	80.9 ± 8.7	0.413
Bands, %	2.7 ± 6.3	2.5 ± 4.4	0.792
CRP, mg/L	125 ± 95.6	188.3 ± 105.8	<0.001
Management			
Emergency appendectomy	106 (100.0%)	80 (78.4%)	<0.001
Interval appendectomy	0 (0.0%)	22 (21.6%)	<0.001
Outcome/complications			
Length of stay in hospital (days)	7.2 ± 4	15.4 ± 6.5	<0.001

Data given as means ± standard deviation or numbers (percentages). RLQ = right lower quadrant; WBC = white blood count; CRP = C-reactive protein; ns =no statistical significance was set at *p* > 0.05. *p*-value was determined by Student’s *t*-test or chi-square test when appropriate.

**Table 2 ijms-25-06076-t002:** Comparison of IL-6 levels in children with non-perforated and perforated appendices from days 1 to 3.

	IL-6 (pg/mL)		
Duration	Non-Perforated Appendices	Perforated Appendices	*p*-Value
	Mean ± SD (N)	Mean ± SD (N)	
Day 1	146 ± 184.34 (27)	143.24 ± 200.45 (6)	0.469
Day 2	112.90 ± 102.84 (26)	254.44 ± 407.60 (30)	0.012
Day 3	117.27 ± 66.97 (28)	1111.44 ± 1709.16 (22)	<0.001

**Table 3 ijms-25-06076-t003:** Comparison of TNF-α levels in children with non-perforated and perforated appendices from days 1 to 3.

	TNF-α (pg/mL)		
Duration	Non-Perforated Appendices	Perforated Appendices	*p*-Value
	Mean ± SD (N)	Mean ± SD (N)	
Day 1	105.62 ± 12.14 (23)	189.78 ± 91.47 (6)	0.020
Day 2	111.60 ± 30.47 (22)	228.05 ± 167.24 (29)	<0.001
Day 3	143.20 ± 62.92 (22)	226.67 ± 254.51 (25)	0.179

**Table 4 ijms-25-06076-t004:** Comparison of ICAM-1 levels in children with non-perforated and perforated appendices from days 1 to 3.

	ICAM-1 (pg/mL)		
Duration	Non-Perforated Appendices	Perforated Appendices	*p*-Value
	Mean ± SD (N)	Mean ± SD (N)	
Day 1	78.69 ± 148.79 (32)	959.71 ± 121.78 (8)	<0.001
Day 2	423.21 ± 531.77 (48)	936.96 ± 288.43 (64)	<0.001
Day 3	402.11 ± 478.47 (24)	1305.19 ± 344.13 (30)	<0.001

**Table 5 ijms-25-06076-t005:** Comparison of E-selectin levels in children with non-perforated and perforated appendices from days 1 to 3.

	E-Selectin (pg/mL)		
Duration	Non-Perforated Appendices	Perforated Appendices	*p*-Value
	Mean ± SD (N)	Mean ± SD (N)	
Day 1	27.54 ± 25.04 (23)	152.51 ± 39.26 (8)	<0.001
Day 2	170.52 ± 299.62 (42)	246.27 ± 216.16 (65)	<0.001
Day 3	157.73 ± 202.49 (20)	466.34 ± 287.02 (30)	<0.001

## Data Availability

The datasets used and/or analyzed during the current study are available from the corresponding author upon reasonable request.

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
