# Peer review of "Changes in Levels of Serum Cytokines and Chemokines in Perforated Appendicitis in Children"

_ijms, 2024, doi:10.3390/ijms25116076_

Round 1

Reviewer 1 Report

Comments and Suggestions for Authors

1.        The abstract begins with a statement without contextualization that should be reformulated

2.        Rather than speaking of ruptured appendix, it is more appropriate to speak of advanced/complicated and uncomplicated/uncomplicated appendicitis or appendiceal perforation, in more academic terms.

3.        In the abstract the authors do not specify whether the samples obtained between days 1 and 3 are all pre-surgical or post-surgical.

4.        In the introduction the authors omit specific recent references regarding the role of IL-6 in the distinction between complicated and uncomplicated acute appendicitis in children and its prognostic role in pediatric acute appendicitis. I recommend reviewing.

Tegarding the role of IL-6 in pediatric acute appendicitis.

Arredondo Montero J, Bardají Pascual C, Bronte Anaut M, López-Andrés N, Antona G, Martín-Calvo N. Diagnostic performance of serum interleukin-6 in pediatric acute appendicitis: a systematic review. World J Pediatr. 2022 Feb;18(2):91-99. doi: 10.1007/s12519-021-00488-z. Epub 2022 Jan 3. PMID: 34978051.

Zhang T, Cheng Y, Zhou Y, Zhang Z, Qi S, Pan Z. Diagnostic performance of type I hypersensitivity-specific markers combined with CRP and IL-6 in complicated acute appendicitis in pediatric patients. Int Immunopharmacol. 2023 Nov;124(Pt B):110977. doi: 10.1016/j.intimp.2023.110977. Epub 2023 Sep 27. PMID: 37774482.

Regarding the discriminative capacity of IL-6 between complicated and uncomplicated acute appendicitis.

Arredondo Montero J, Antona G, Rivero Marcotegui A, Bardají Pascual C, Bronte Anaut M, Ros Briones R, Fernández-Celis A, López-Andrés N, Martín-Calvo N. Discriminatory capacity of serum interleukin-6 between complicated and uncomplicated acute appendicitis in children: a prospective validation study. World J Pediatr. 2022 Dec;18(12):810-817. doi: 10.1007/s12519-022-00598-2. Epub 2022 Sep 16. PMID: 36114365; PMCID: PMC9617836.

Kakar M, Delorme M, Broks R, Asare L, Butnere M, Reinis A, Engelis A, Kroica J, Saxena A, Petersons A. Determining acute complicated and uncomplicated appendicitis using serum and urine biomarkers: interleukin-6 and neutrophil gelatinase-associated lipocalin. Pediatr Surg Int. 2020 May;36(5):629-636. doi: 10.1007/s00383-020-04650-y. Epub 2020 Mar 26. PMID: 32219562.

Elliver M, Salö M, Roth B, Ohlsson B, Hagander L, Gudjonsdottir J. Associations between Th1-related cytokines and complicated pediatric appendicitis. Sci Rep. 2024 Feb 26;14(1):4613. doi: 10.1038/s41598-024-53756-z. PMID: 38409170; PMCID: PMC10897334.
Regarding the post-surgical prognostic role of IL-6 in pediatric acute appendicitis.

Latifi SQ, O'Riordan MA, Levine AD, et al. Persistent elevation of serum interleukin-6 in intraabdominal sepsis identifies those with prolonged length of stay. J Pediatr Surg 2004;39(10):1548–1552;

Arredondo Montero J, Rivero Marcotegui A, Bardají Pascual C, Antona G, Fernández-Celis A, López-Andrés N, Martín-Calvo N. Post-Operative Increase in Serum Interleukin-6 Is Associated With Longer Hospital Stay in Children Operated on for Acute Appendicitis: A Pilot Study. Surg Infect (Larchmt). 2023 Sep;24(7):619-624. doi: 10.1089/sur.2023.091. PMID: 37695684.

5.     “when an uninflamed appendix was found in 96 a patient who had undergone surgery (a normal appendectomy).” Did the authors rely on intraoperative or histological criteria to classify an appendix as “uninflammed”?

6.     The manuscript needs an extensive English revision by a native speaker.

7.     If the specimens were divided into SA, GA and RA, does this mean that the authors did not have any negative appendectomy (appendix without inflammation?).

8.     Authors use abbreviations without prior clarification (WBC, CRP).

9.     In statistical methodology, the authors do not specify whether the normality of quantitative variables was evaluated. Was the variance analyzed?  Why are nonparametric tests presented but the data are presented as mean and standard deviation instead of median and interquartile range? This manuscript needs review by an expert statistician.

10.   I do not understand the concept of “normal appendicitis.”

11.   Putting ns in a statistical analysis of a scientific paper is not acceptable. It is not the same a p-value of 0.06 than a p-value of 0.97. Please report all two-tailed p-values for each bivariate analysis.

12.   Why the authors have not created three groups for the analysis of table 1 (SA, GA and RA) and have done 3-band analysis (Kruskal-wallis)? Combining the GA and SA group is scientifically debatable.

13.   It is striking the length of hospitalization reported by the authors is. Any comments?

14.   Classifying interval appendectomies histologically as perforated appendices is highly debatable. Were the authors based on radiology to classify as perforated appendix? An interval appendectomy 3-6 months after the event does not usually present microscopic perforation...clarify this.

15.   Previous work by Arredondo et al. reported the wide analytical range of IL-6 in patients with pediatric appendicitis (especially complicated). Please see previous attached references. It would be interesting to comment on the issue of outliers and their applicability in clinical practice. There is a systematic review on calprotectin and the APPY-1 test in pediatric appendicitis that also comments on this aspect. I recommend reviewing it.

16.   As previously mentioned, the analyses in Table 2 for day 1 have marginal significance (0.469) because the SA and GA groups have been pooled. Comment.

17.   Interestingly, and in the analyses of my working group, ICAM-1 did not reach statistical significance in any of the analyses reported by the authors. I think it would be interesting to delve a little deeper into the values obtained and why the authors chose this molecule.

18.   “up to 20% of children undergoing appendectomy have postoperative complications” I absolutely disagree. The reference used is limited to a retrospective Spanish series of low sample size in a journal not indexed in JCR. Look for better quality references.

Comments on the Quality of English Language

 The manuscript needs an extensive English revision by a native speaker.

Reviewer 2 Report

Comments and Suggestions for Authors

The authors present a prospective study on the value of preoperative measurement of several proinflammatory biomarkers in the diagnosis of perforated appendicitis in the pediatric population. The topic is of high clinical importance and therefore any additional insights are potentially highly benefitial. The authors show that increased levels of IL-6, TNF-alpha, ICAM-1 and E-selectin exist in appencitis patients and may serve as helpful parameters in the prediction of appendicitis in children.

Major comments:

 1.       Methods: It is not clear, from how many of the patients specimens of all three days were available. Since the study design did not change the therapeutic pathway of the patients (observational approach), surely not all of the patients were 3 days in the hospital before they were operated on. Please specify which proportion of the patients were operated on day 1, day 3 and day 3 and were there any differences between the groups (e.d. higher biomarker values in patients who were operated on on day 1, possibily indicating rapidly progressing disease).

2. Results: The total number of patients is confusing: 226 patients  with lower RLQ were admitted to the ED. All of these (?) underwent surgical intervention. Probably not, but please specify which proportion of the probably higher number of patients were included in the study.

3.       There must be proinflammatory biomarker results from patients having no appendicitis (either being not operated on after three days at the latest) and discharged with regredient symptoms or from the 18 patients with histological normal appendicitis. How were the values in these patients compared to SA and RA? If the biomarkers tested in the study are of clinical value they should also be able to distinguish between normal appendicitis and appendicitis.

4.       The only other laboratory marker which was significantly different between the two groups was CRP. How was correlation between CRP and the biomarkes on day 1,2,3, respectively? Did they e.g. show a quicker or stronger increase in RA patients?`This would also add value to the findings that these biomarkers are of clinical value.

5.       For clinical use, sensivity and specificity calculataion would be of greates meaningfulness: Are there any cutoffs (values of these biomarkers) to distinguish between SA and RA, for example table 3 is suggestive of a rapid increase of TNF-alpha on day 2. If a TNF-alpha of, let’s say as an example 150 pg/ml on day 2 is used, how sensitive was it to diagnose RA?

6.       Conclusion: The authors conclude that the biomarkers used in the study could be used initially in the pre-operative prediction of perforated appendicitis. This is still a bit far-fetched. I recommend rephrasing towards a more careful wording, especially if there are no sensivitiy/specifity calculations.

Minor comments:

The authors use the term “ruptured appendicitis” throughout the paper, but the term “perforated appendicitis” (which is the better term) in the title. Please align or consider using the term “complicated appendicitis” as opposed to “simple appendicitis”.

Discussion, second paragraph: Omit the sentence “The radiologist may be uncertain….” – This sentence only explains the term “equivocal” and is not necessary.

Results section, line 142. “Of” instead of “of” (new sentence). See also major comment no. 2. Possibly rephrase.

Otherwise the paper is well written, language is fine and the discussion is appropriate.

Round 2

Reviewer 1 Report

Comments and Suggestions for Authors

The manuscript has been improved by the changes introduced. 

Comments on the Quality of English Language

A moderate revision of English is recommended. 

Reviewer 2 Report

Comments and Suggestions for Authors

Wording has been updated throughout the manuscript and discussion has been updated. The manuscript has improved thoroughly throughout the process.